# Atlas of voluntary facial muscle activation: Visualization of surface electromyographic activities of facial muscles during mimic exercises

Nikolaus P. Schumann[1], Kevin Bongers[1], Hans C. Scholle[1], Orlando Guntinas-Lichius[2]*

1 Division Motor Research, Pathophysiology and Biomechanics, Department of Trauma, Hand and Reconstructive Surgery, Jena University Hospital, Friedrich-Schiller-University Jena, Jena, Germany,
2 Department of Otolaryngology, Jena University Hospital, Friedrich-Schiller-University Jena, Jena, Germany

* Orlando.guntinas@med.uni-jena.de

**Data Availability Statement:** All relevant data are within the paper and its Supporting Information files.

## Abstract

Complex facial muscle movements are essential for many motoric and emotional functions. Facial muscles are unique in the musculoskeletal system as they are interwoven, so that the contraction of one muscle influences the contractility characteristic of other mimic muscles. The facial muscles act more as a whole than as single facial muscle movements. The standard for clinical and psychosocial experiments to detect these complex interactions is surface electromyography (sEMG). What is missing, is an atlas showing which facial muscles are activated during specific tasks. Based on high-resolution sEMG data of 10 facial muscles of both sides of the face simultaneously recorded during 29 different facial muscle tasks, an atlas visualizing voluntary facial muscle activation was developed. For each task, the mean normalized EMG amplitudes of the examined facial muscles were visualized by colors. The colors were spread between the lowest and highest EMG activity. Gray shades represent no to very low EMG activities, light and dark brown shades represent low to medium EMG activities and red shades represent high to very high EMG activities relatively with respect to each task. The present atlas should become a helpful tool to design sEMG experiments not only for clinical trials and psychological experiments, but also for speech therapy and orofacial rehabilitation studies.

## Introduction

Facial movements by contraction of facial muscles support manifold functions in human behavior [1]. They participate in automatic somatic and visceral motor programs. They are important to reflect emotions, display current mood, and are essential for non-verbal communication. The facial muscular system is composed of a flat web of muscular fascicles. They are embedded in a 2-dimensional space and form a complex interdependent system that is connected to the skin [1]. Consequently, facial muscle contractions change the superficial geometry of the face. This is unique and not the only difference to other human muscles. They have

**Funding:** Orlando Guntinas-Lichius acknowledges support by a Deutsche Forschungsgemeinschaft (DFG) grant GU-463/12-1.

**Competing interests:** The authors have declared that no competing interests exist.

no fixed insertion points, are interwoven, partly overlapping, and isolated activation of a single muscle is the exception. Nevertheless, anatomical textbooks typical classify facial muscles as individual muscles with individual function [2].

Surface facial electromyography (sEMG) represents an appropriate psychophysiological measure to test group activity of some facial muscles and its association to specific emotions [3]. Furthermore, needle EMG is a standard method for diagnostics in patients with facial nerve dysfunction [4]. sEMG is less frequently used for the assessment of patients with facial nerve dysfunction [5]. Facial EMG recordings using only two to six pairs needle or surface electrodes during the resting state or voluntary movement cannot acquire complete information on the complex activation of facial muscles [6]. Fridlund and Cacioppo therefore recommended for psychophysiological experiments the sEMG recordings from ten facial muscles [7]. Recently, Kuramoto et al. recommended the use of 24 sEMG electrodes in an EEG-like arrangement over the facial independent of the underlying muscles and a myogenic potential topogram analysis [8]. Experimental high-density sEMG (HD sEMG) with 90 electrodes even allows a description of facial muscle activation with activation maps projected on the facial surface [6].

Some years ago, we published a study on facial muscle activation patterns based on multi-channel sEMG [9]. This high-resolution sEMG analysis showed distinct task-specific interactions between individual facial muscles. The mean sEMG amplitude characteristics for each facial muscle were specifically dependent on the performed activation task. Since this publication, we are frequently asked if it would be possible to visualize these task-specific facial activation patterns. This would help to define sEMG recording patterns for psychological experiments as well as for clinical studies. This is the reason why we know transposed the sEMG amplitude characteristics in a color-coded atlas of voluntary facial muscle activation tasks.

## Materials and methods

### Subjects

The data set of a previous study was used [9]. Briefly, this data set included 30 healthy male volunteers with no neurological diseases (mean age: 26± 3.2 years) All subjects gave written informed consent to participate in the study. The ethics committee of the Jena University Hospital approved the study (No. 2129-10/07). The individual shown in the figures of this manuscript has given written informed consent to publish these case details.

### Facial surface electromyography (sEMG) registration

The registration technique is described in detail elsewhere [9]. For this new study we used the 48-channel sEMG data recorded in the already mentioned previous study [9]. Briefly, surface electrodes were used (Ag-AgCl discs, diameter of 4 mm, Zentner, Freiburg, Germany). The reference electrodes were set at the ear lobes and a ground electrode at the mastoid. The monopolar sEMG recording was performed with a multi-channel EMG system (3 dB level frequency range of the EMG amplifiers: 10–700 Hz; sampling rate 3000/s; resolution: 2.44 μV/bit; Biovision, Wehrheim, Germany). Monopolar electromyograms were recorded simultaneously from both sides of the face. The following ten facial muscles were recorded at the same time: frontalis, orbicularis oculi, zygomatic major, zygomatic minor, levator labii superioris/levator labii superioris alaeque nasi, orbicularis oris, depressor anguli oris, depressor labii, and mentalis muscle. Furthermore, both external chewing muscles, the masseter and temporal muscle were recorded for control reasons. **Fig 1** shows the electrode positions.

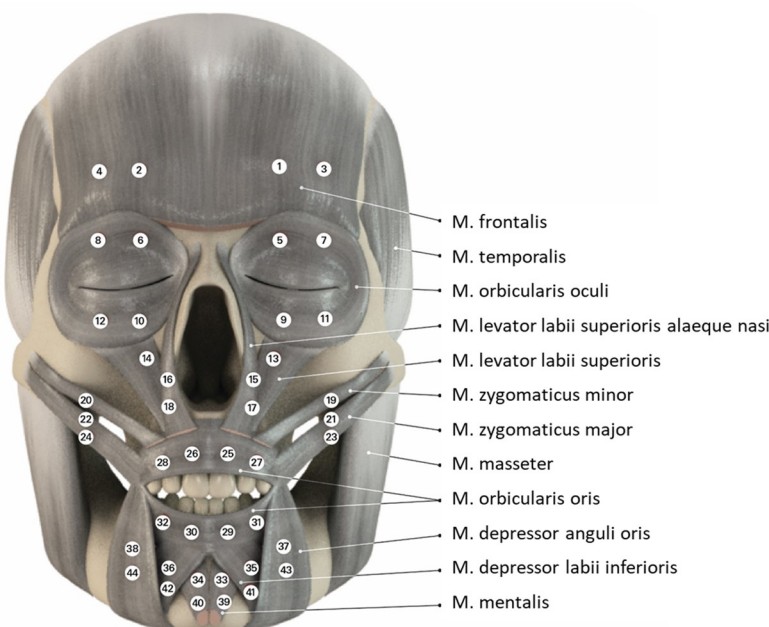

**Fig 1. sEMG electrode arrangement.** 44 electrodes were placed symmetrically on 10 mimic muscles on both sided of the face, respectively.

## Mimic exercises

The facial motor tasks and the muscles with the highest sEMG activity in each tasks are summarized in **Table 1**. In the six tasks comprised the pronunciation of the six German vowels. Subsequently, typical mimic expressions were performed related to all parts of the face. Some tasks were performed unilaterally. The participants sat in relaxed upright position during the exercises, face to face with the examiner. The examiner named and demonstrated each planned facial movement. The participants tried the exercise first. The examiner corrected the participant if needed.

## Color coding for visualization of the relative sEMG activity

A professional graphic designer (see acknowledgment) visualized the facial skull and the overlying facial muscles in 3D based on the description of the facial muscles in an anatomy textbook [10] using a 3D visualization software (Maya, Autodesk Inc., San Rafael, United States). The hue, saturation, value (HSV) color coding reached from white (no activation) over orange to red (maximal activation). The mean normalized EMG amplitudes of the examined facial muscles were visualized by the HSV colors determined by linear interpolation. The statistics for the normalization process are published elsewhere [9]. In order to achieve an optimal resolution of the EMG activity pattern, a separate color scale was adapted to each of the 29 facial movement tasks. The colors were spread between the lowest and highest EMG activity. Since the minimal values the EMG activities hardly differed between the facial movement tasks, the range of the color scale was determined by the highest EMG activity value (maximum of the mean EMG amplitude). Gray shades represent no to very low EMG activities, light and dark brown shades represent low to medium EMG activities and red shades represent high to very high EMG activities, relatively with respect to each task.

**Table 1. Overview about the mimic tasks.**

| No. | Description of the task (T) | Muscles with highest sEMG activity |
|---|---|---|
| 1 | Pronouncing the German vowel: A/aː/* | inferior orbicularis oris, mentalis, depressor labii |
| 2 | Pronouncing the German vowel: Ä/æ/* | inferior orbicularis oris, depressor labii, mentalis |
| 3 | Pronouncing the German vowel: E/eː/* | inferior orbicularis oris, depressor labii, mentalis |
| 4 | Pronouncing the German vowel: I/iː/* | inferior orbicularis oris, depressor labii, mentalis |
| 5 | Pronouncing the German vowel: O/oː/* | inferior orbicularis oris, depressor labii |
| 6 | Pronouncing the German vowel: U/uː/* | inferior orbicularis oris, depressor labii |
| 7 | Pressing lips together | orbicularis oris, depressor anguli oris, mentalis, depressor labii, zygomatic |
| 8 | Pulling corners of the mouth downwards | inferior orbicularis oris, depressor anguli oris, depressor labii, mentalis |
| 9 | Voluntary smiling: Pulling corners of the mouth upwards and backwards | zygomatic, orbicularis oris, mentalis, depressor labii, depressor anguli oris |
| 10 | Depressing lower lip | inferior orbicularis oris, depressor labii, mentalis |
| 11 | Protruding lower lip | mentalis, depressor anguli oris, depressor labii, orbicularis oris |
| 12 | Pulling upper lip upwards | levator labii superioris alaeque nasi, levator labii superioris, orbicularis oris |
| 13 | Pulling upper lip upwards and depressing lower lip simultaneously | depressor labii, inferior orbicularis oris, mentalis |
| 14 | Pursing lips | orbicularis oris |
| 15 | Blowing out cheeks | orbicularis oris, mentalis, depressor labii |
| 16 | Sucking cheeks inward | orbicularis oris, mentalis, depressor labii |
| 17 | Whistling with a similar tone pitch | orbicularis oris |
| 18 | Opening jaw with closed lips | mentalis, orbicularis oris, depressor labii, depressor anguli oris |
| 19 | Exhaling forcefully with moderate closed lips | orbicularis oris |
| 20 | Opening lips as wide as possible while the jaw is closed | inferior orbicularis oris, depressor labii, mentalis |
| 21 | Wrinkling the nose | levator labii superioris alaeque nasi |
| 22 | Voluntary smiling only on right side of the face | zygomatic |
| 23 | Voluntary smiling only on left side of the face | zygomatic |
| 24 | Raising eyebrows up and wrinkling the forehead | frontalis |
| 25 | Contracting eyebrows | frontalis |
| 26 | Closing eyelids forcefully | orbicularis oculi, frontalis |
| 27 | Squinting the eyes | orbicularis oculi, frontalis |
| 28 | Closing the right eyelid | right orbicularis oculi, right frontalis, right levator labii superioris |
| 29 | Closing the left eyelid | left orbicularis oculi, left frontalis, left levator labii superioris |

*/international phonetic alphabet/.

## Atlas

The atlas consists of 29 illustrations of the different tasks (Figs 2–30). In the legends for the figures, the distribution from the lowest (minimum) to the highest mean EMG amplitudes (maximum) of each facial movement task is given. In the photos in the upper part of each figure, the respective facial movement is shown.

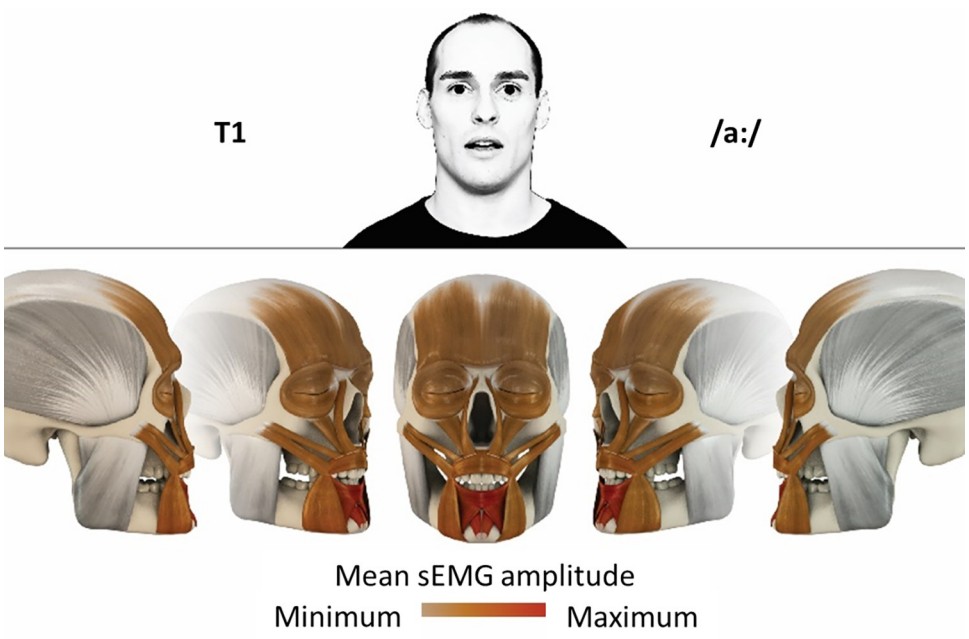

**Fig 2. Color-coded facial muscle activation pattern of task T1.** Pronouncing the German vowel: A/a:/. Animation by Jonas Lauströer.

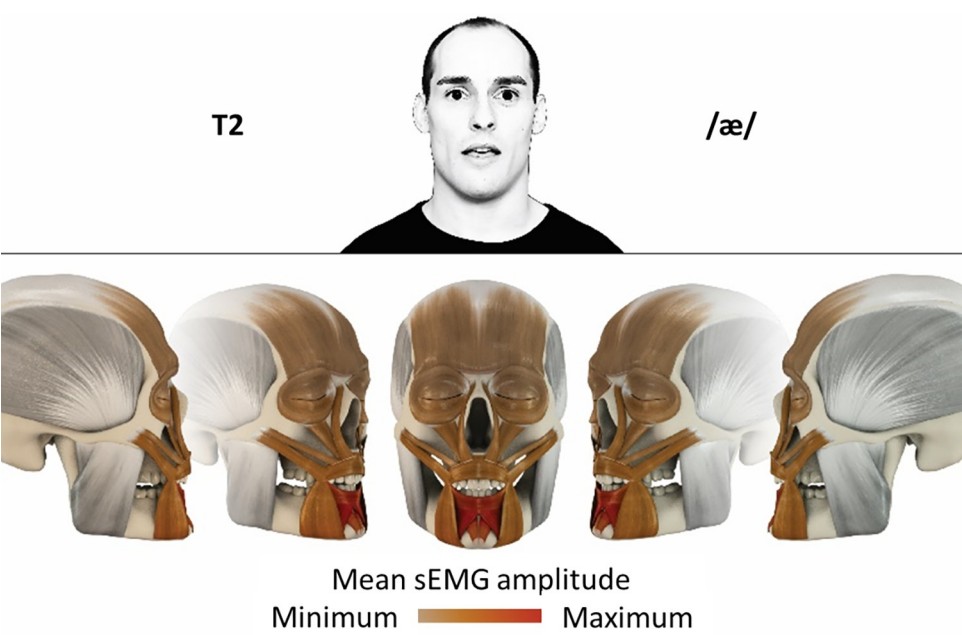

**Fig 3. Color-coded facial muscle activation pattern of task T2.** Pronouncing the German vowel: Ä/æ/. Animation by Jonas Lauströer.

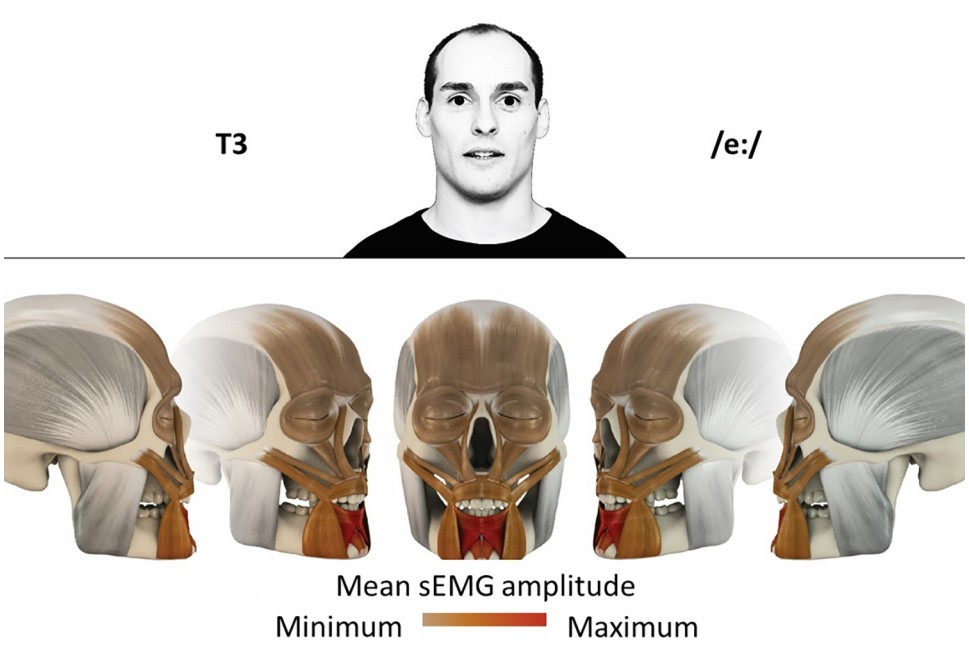

**Fig 4. Color-coded facial muscle activation pattern of task T3.** Pronouncing the German vowel: E/e:/. Animation by Jonas Lauströer.

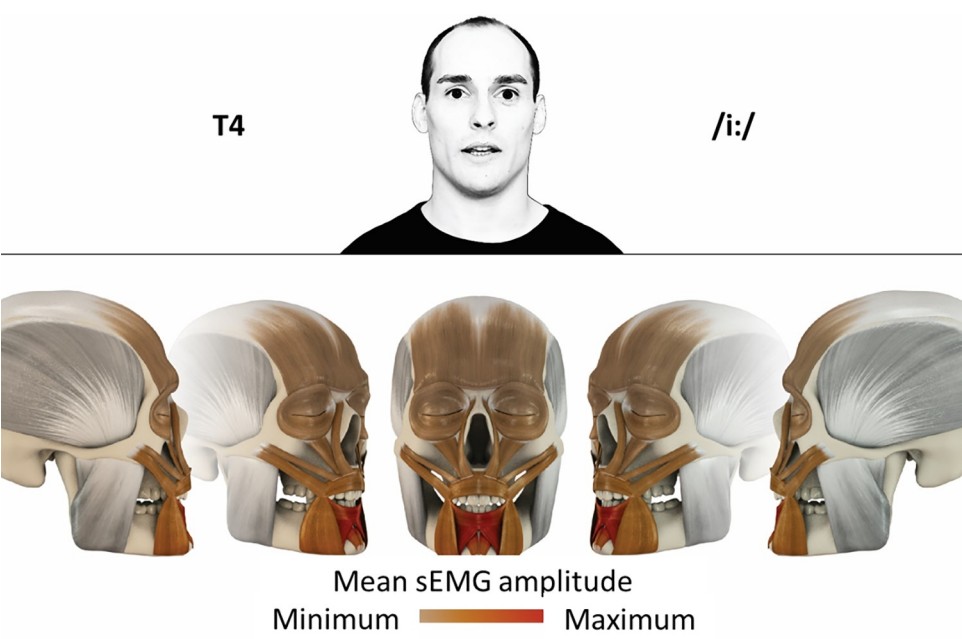

**Fig 5. Color-coded facial muscle activation pattern of task T4.** Pronouncing the German vowel: I/i:/. Animation by Jonas Lauströer.

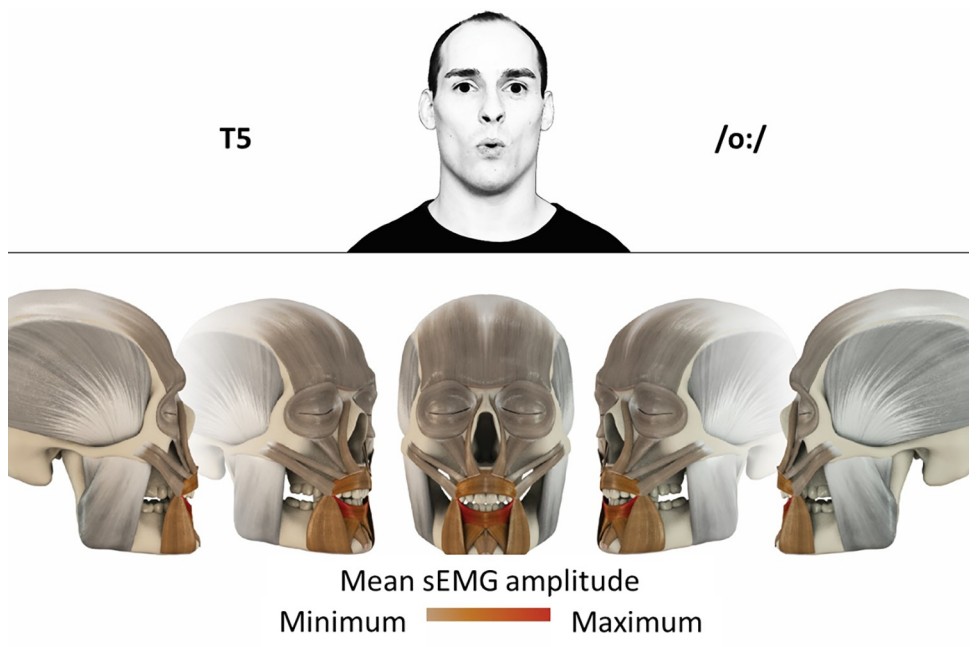

**Fig 6. Color-coded facial muscle activation pattern of task T5.** Pronouncing the German vowel: O/o:/. Animation by Jonas Lauströer.

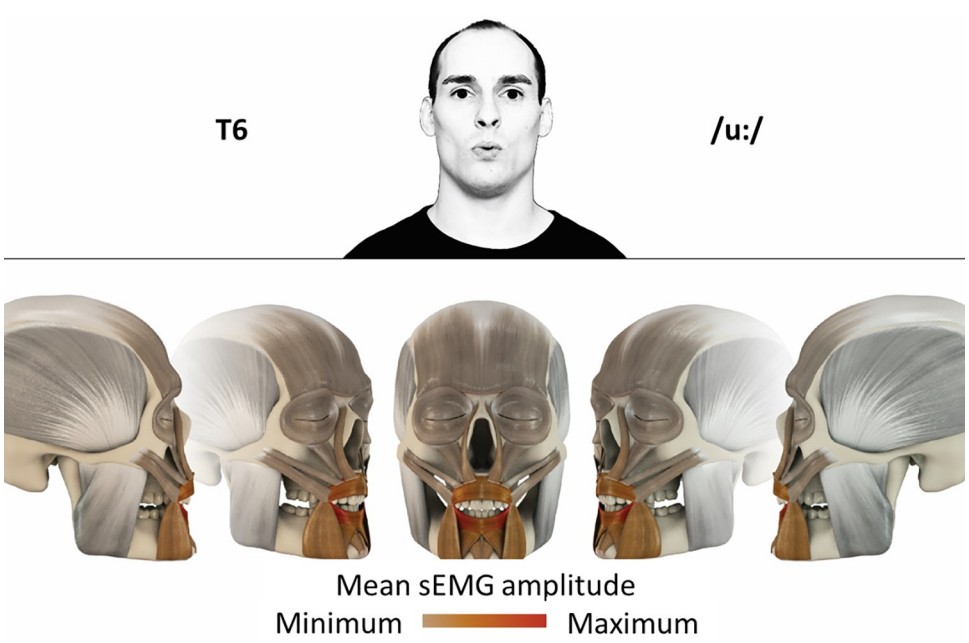

**Fig 7. Color-coded facial muscle activation pattern of task T6.** Pronouncing the German vowel: U/u:/. Animation by Jonas Lauströer.

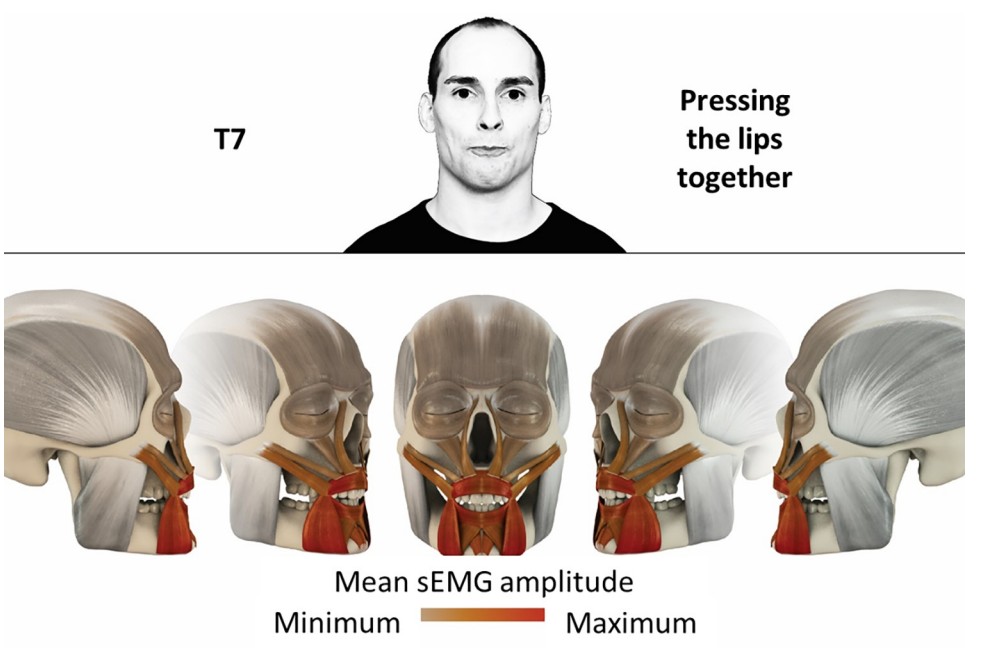

**Fig 8. Color-coded facial muscle activation pattern of task T7.** Pressing lips together.

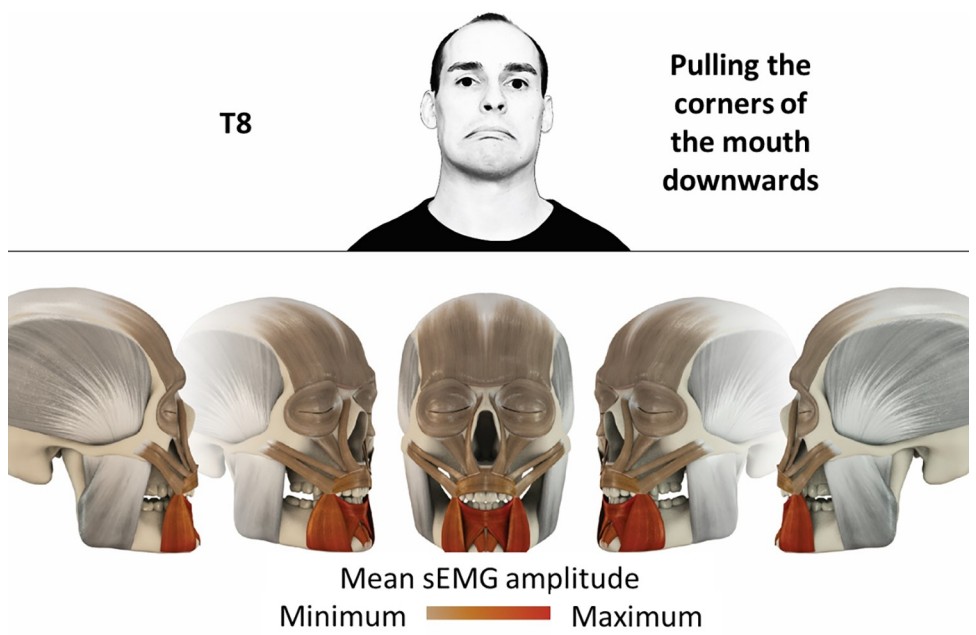

**Fig 9. Color-coded facial muscle activation pattern of task T8.** Pulling corners of the mouth downwards. Animation by Jonas Lauströer.

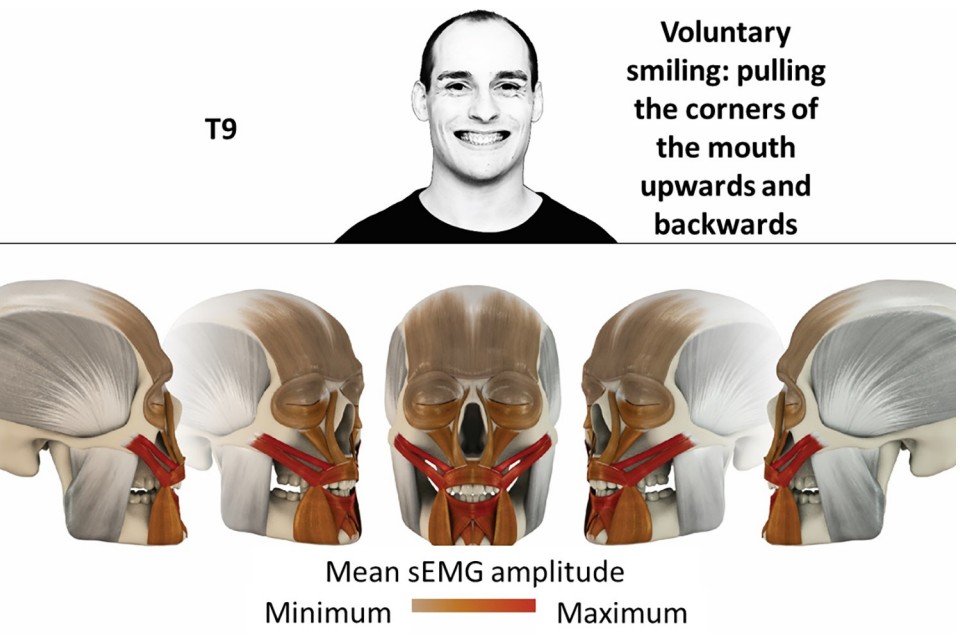

**Fig 10. Color-coded facial muscle activation pattern of task T9.** Voluntary smiling: Pulling corners of the mouth upwards and backwards. Animation by Jonas Lauströer.

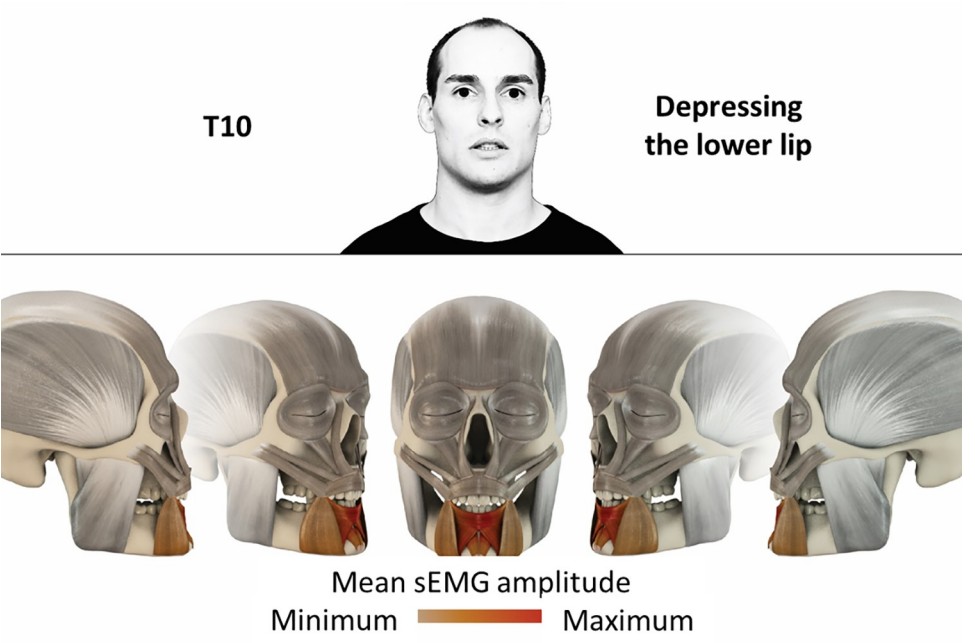

**Fig 11. Color-coded facial muscle activation pattern of task T10.** Depressing lower lip. Animation by Jonas Lauströer.

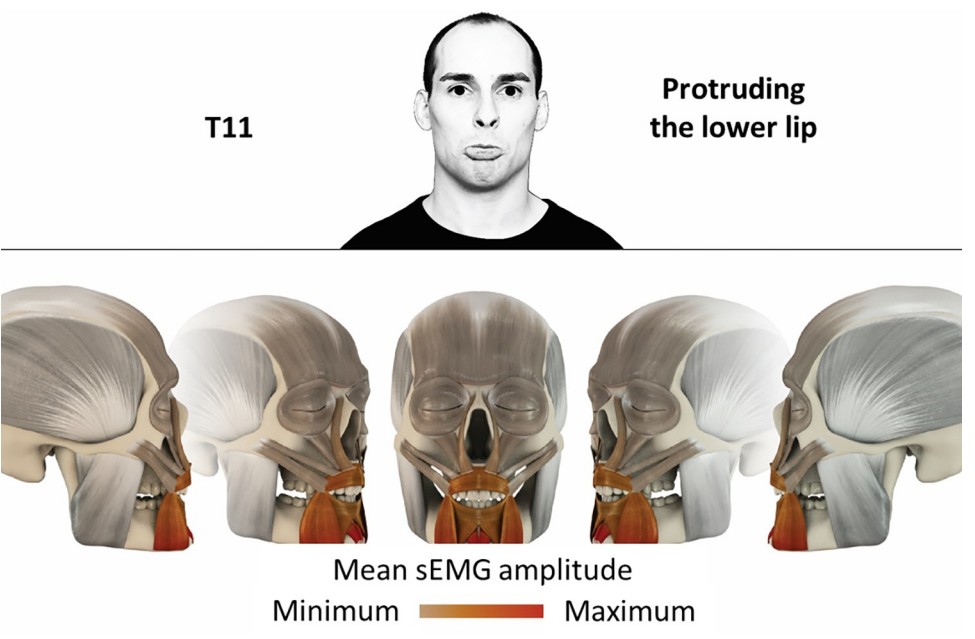

**Fig 12. Color-coded facial muscle activation pattern of task T11.** Protruding lower lip. Animation by Jonas Lauströer.

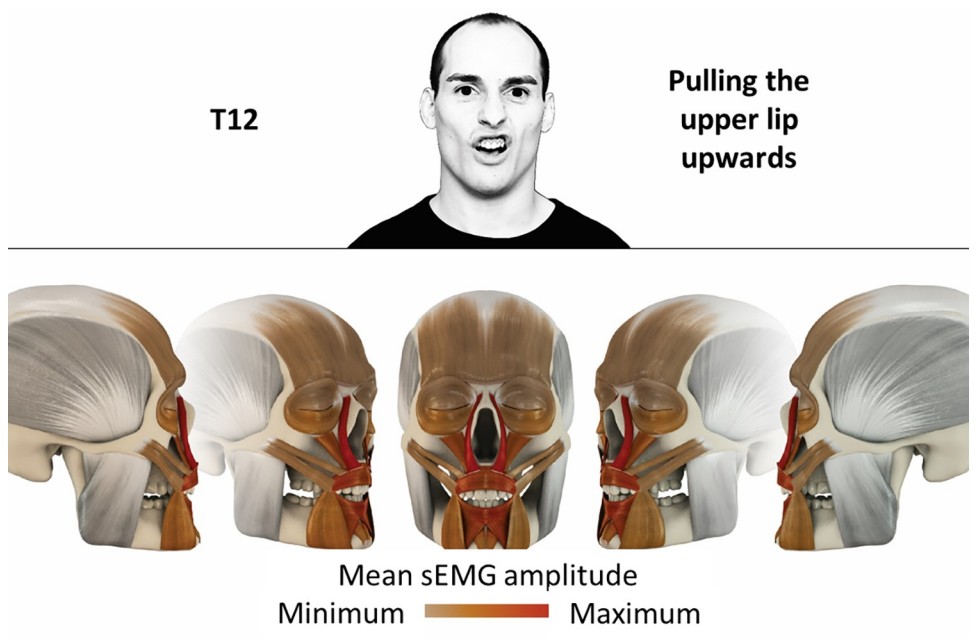

**Fig 13. Color-coded facial muscle activation pattern of task T12.** Pulling upper lip upwards. Animation by Jonas Lauströer.

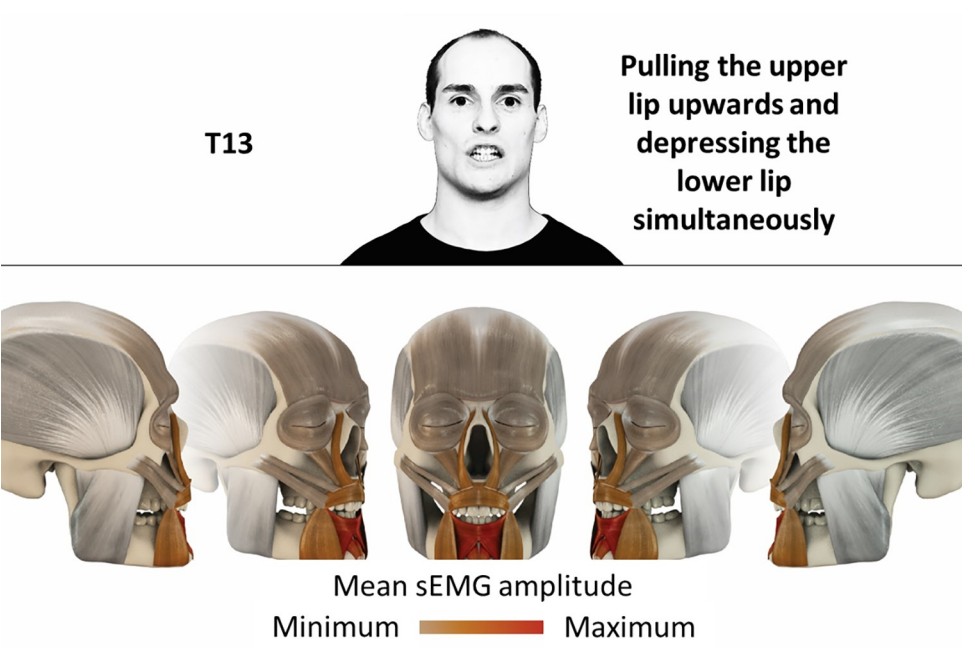

**Fig 14. Color-coded facial muscle activation pattern of task T13.** Pulling upper lip upwards and depressing lower lip simultaneously. Animation by Jonas Lauströer. Animation by Jonas Lauströer.

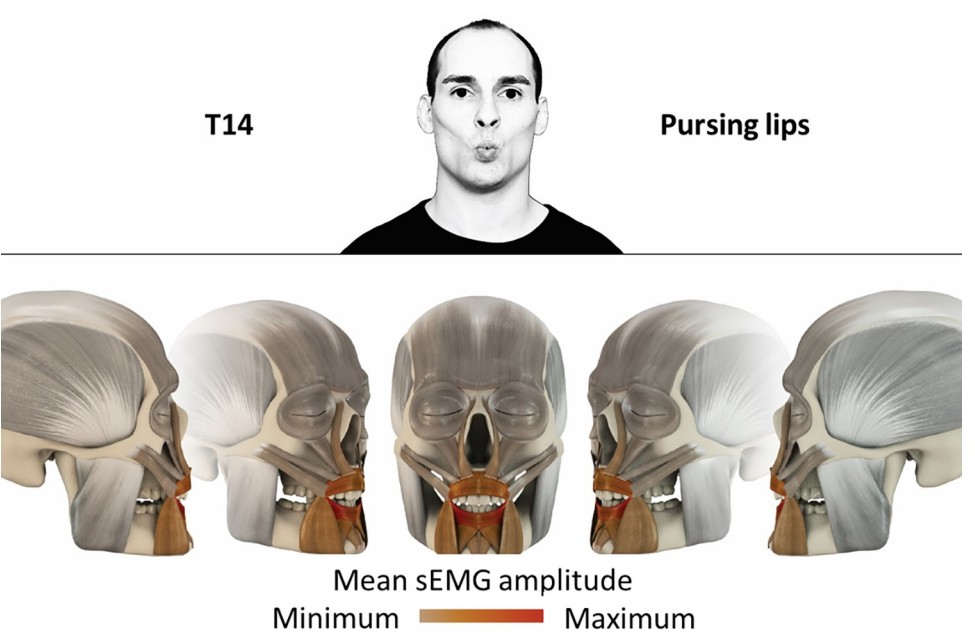

**Fig 15. Color-coded facial muscle activation pattern of task T14.** Pursing lips. Animation by Jonas Lauströer.

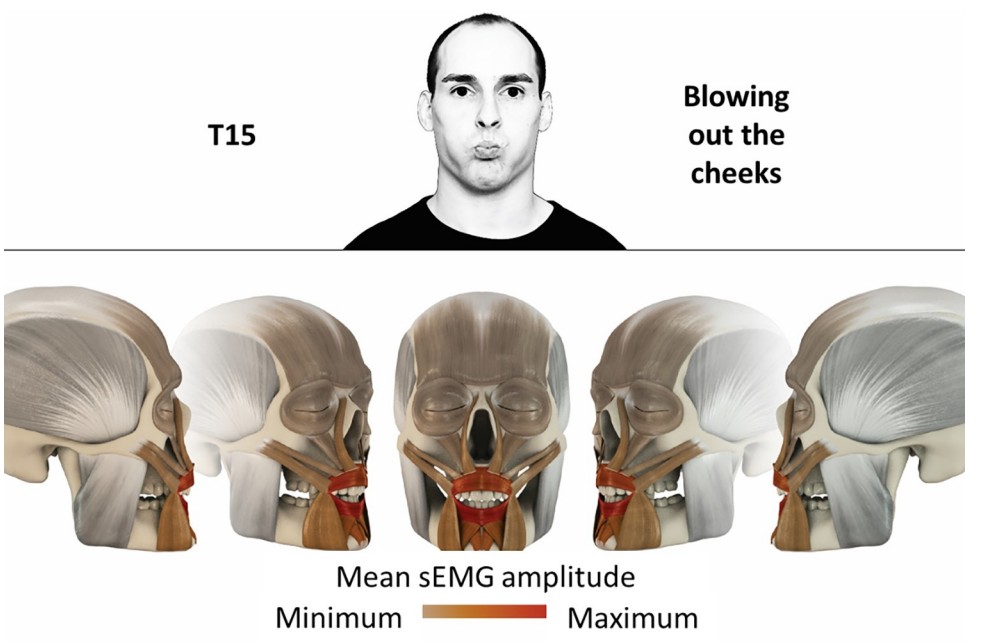

**Fig 16. Color-coded facial muscle activation pattern of task T15.** Blowing out cheeks. Animation by Jonas Lauströer.

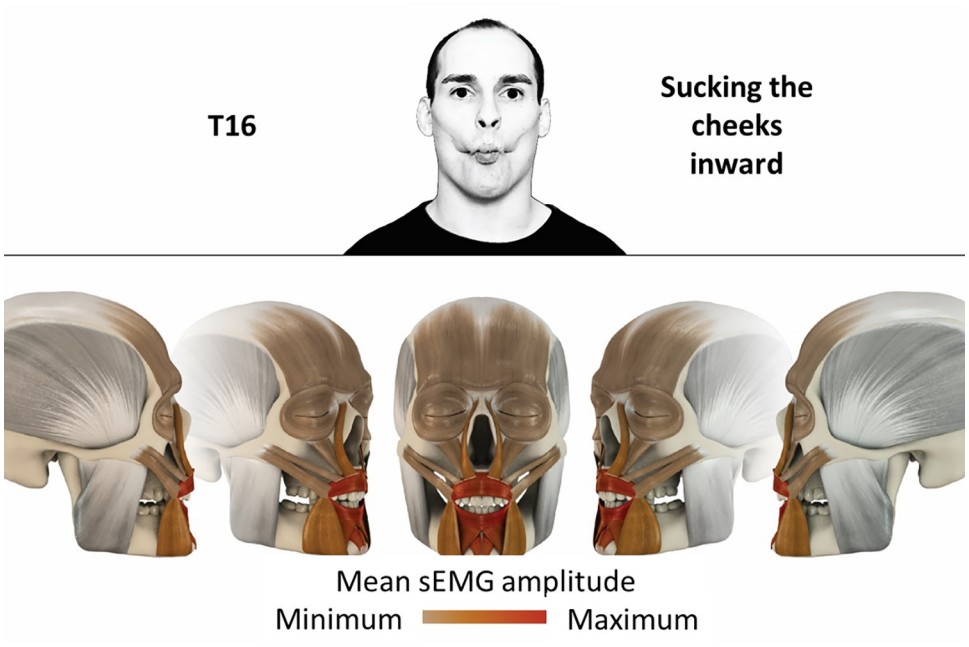

**Fig 17. Color-coded facial muscle activation pattern of task T16.** Sucking cheeks inward. Animation by Jonas Lauströer.

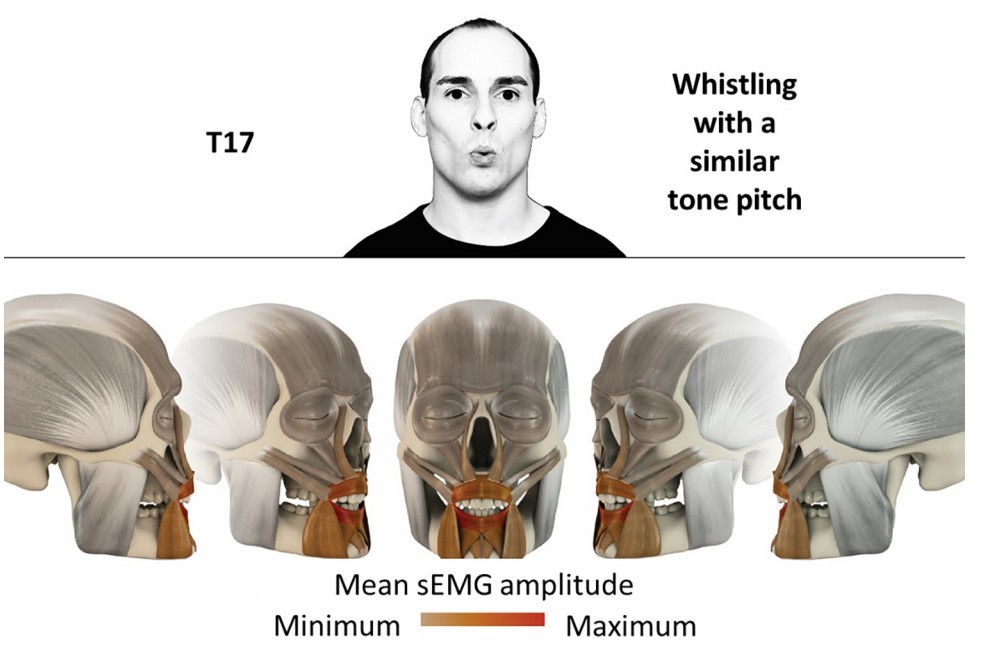

**Fig 18. Color-coded facial muscle activation pattern of task T17.** Whistling with a similar tone pitch. Animation by Jonas Lauströer.

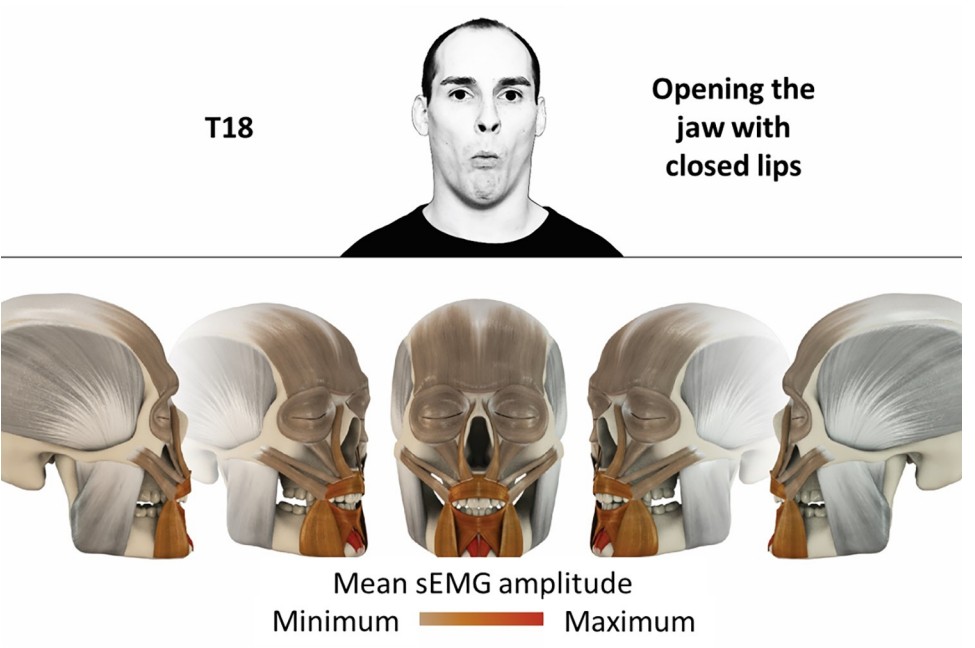

**Fig 19. Color-coded facial muscle activation pattern of task T18.** Opening jaw with closed lips. Animation by Jonas Lauströer.

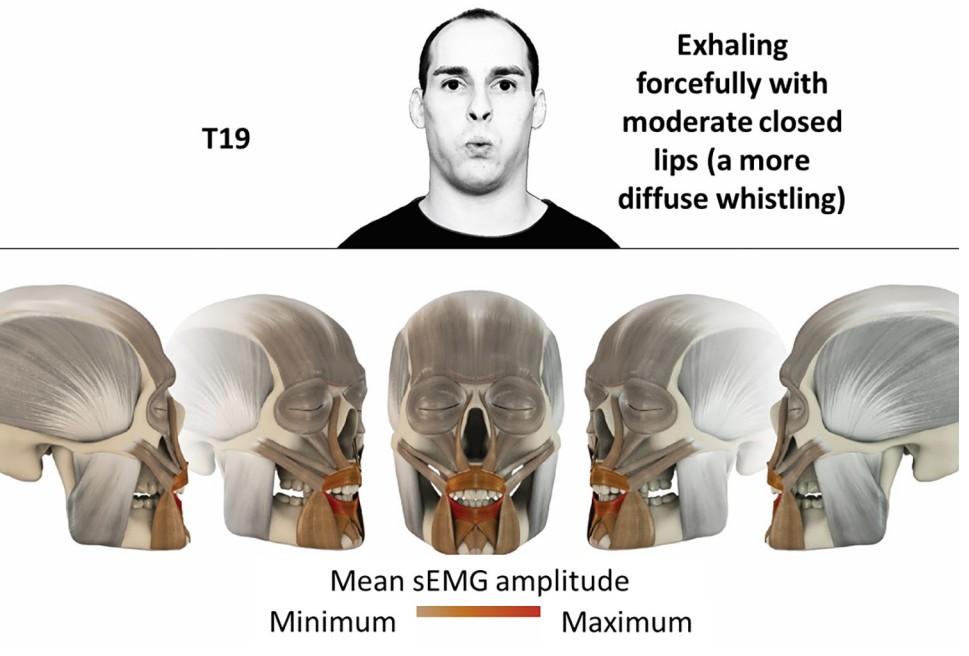

**Fig 20. Color-coded facial muscle activation pattern of task T19.** Exhaling forcefully with moderate closed lips. Animation by Jonas Lauströer.

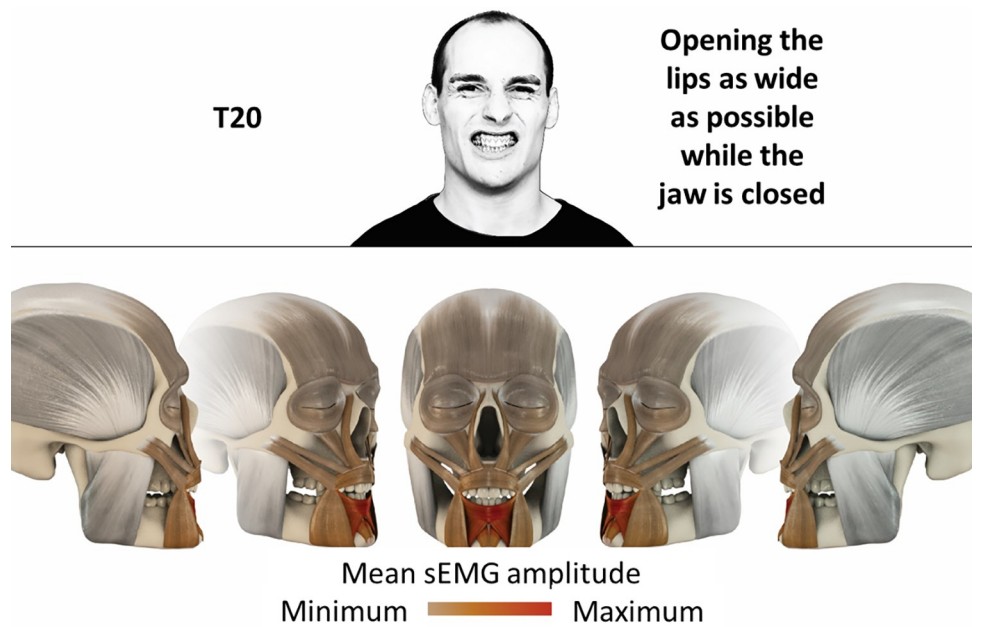

**Fig 21. Color-coded facial muscle activation pattern of task T20.** Opening lips as wide as possible while the jaw is closed. Animation by Jonas Lauströer.

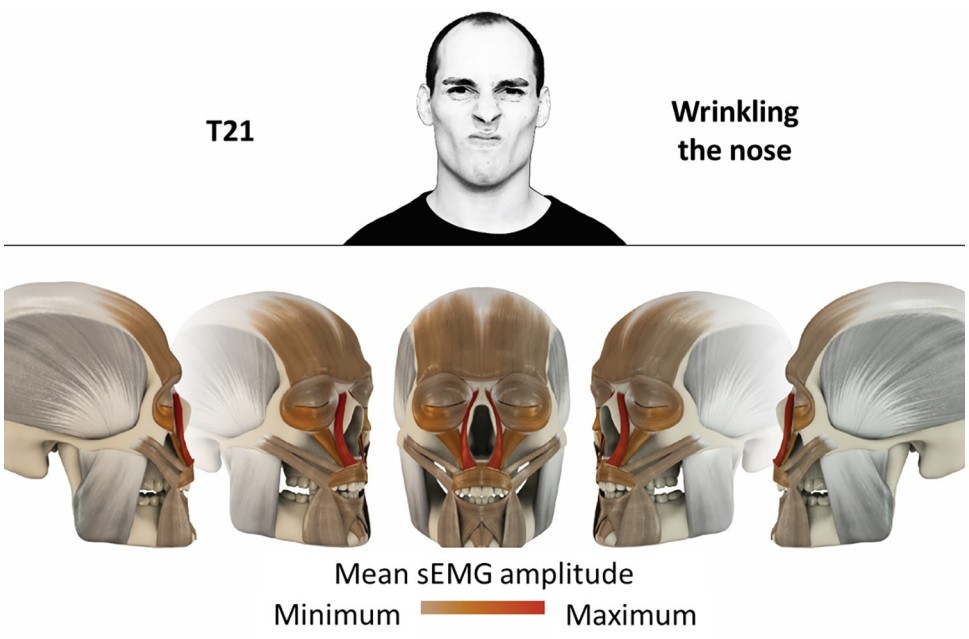

**Fig 22. Color-coded facial muscle activation pattern of task T21.** Wrinkling the nose. Animation by Jonas Lauströer.

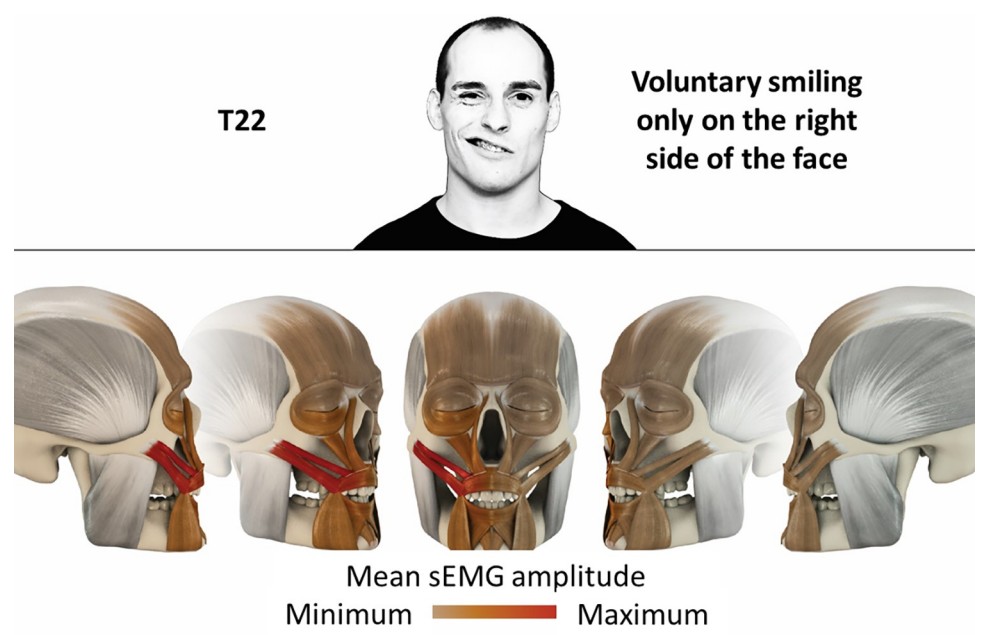

**Fig 23. Color-coded facial muscle activation pattern of task T22.** Voluntary smiling only on right side of the face. Animation by Jonas Lauströer.

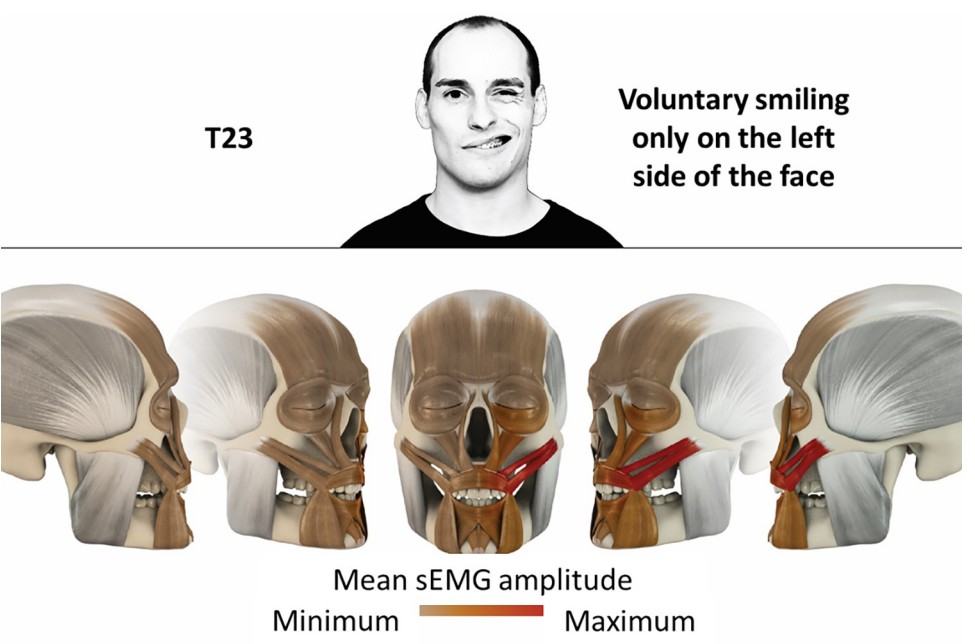

**Fig 24. Color-coded facial muscle activation pattern of task T23.** Voluntary smiling only on left side of the face. Animation by Jonas Lauströer.

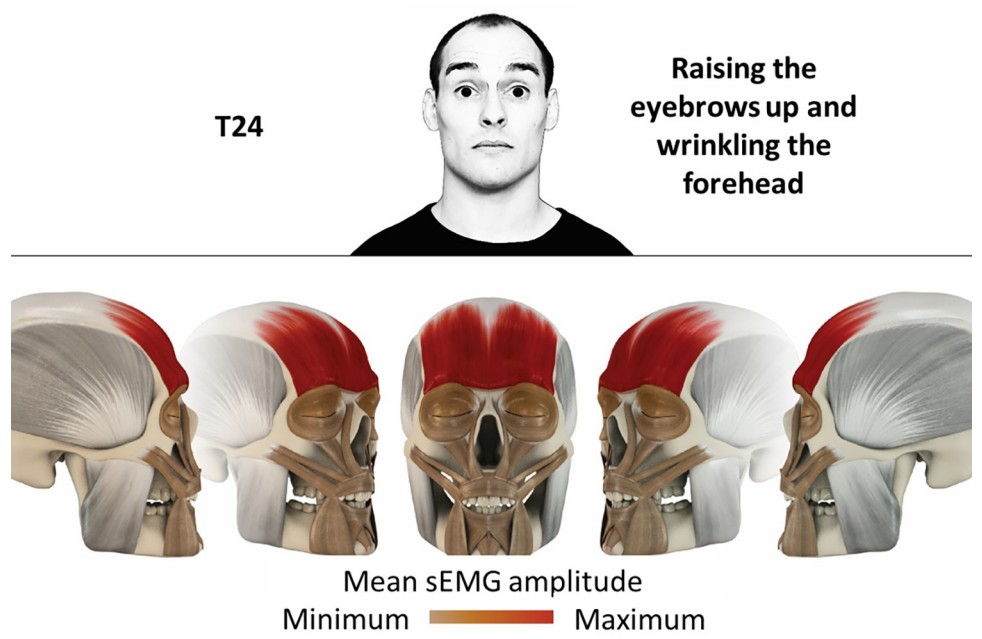

**Fig 25. Color-coded facial muscle activation pattern of task T24.** Raising eyebrows up and wrinkling the forehead. Animation by Jonas Lauströer.

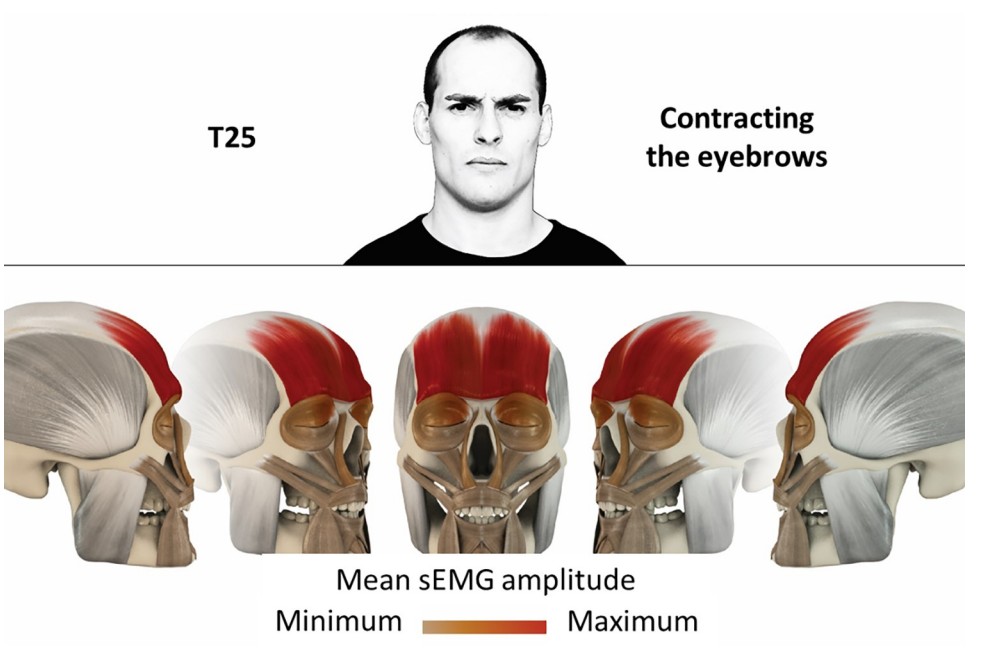

**Fig 26. Color-coded facial muscle activation pattern of task T25.** Contracting eyebrows. Animation by Jonas Lauströer.

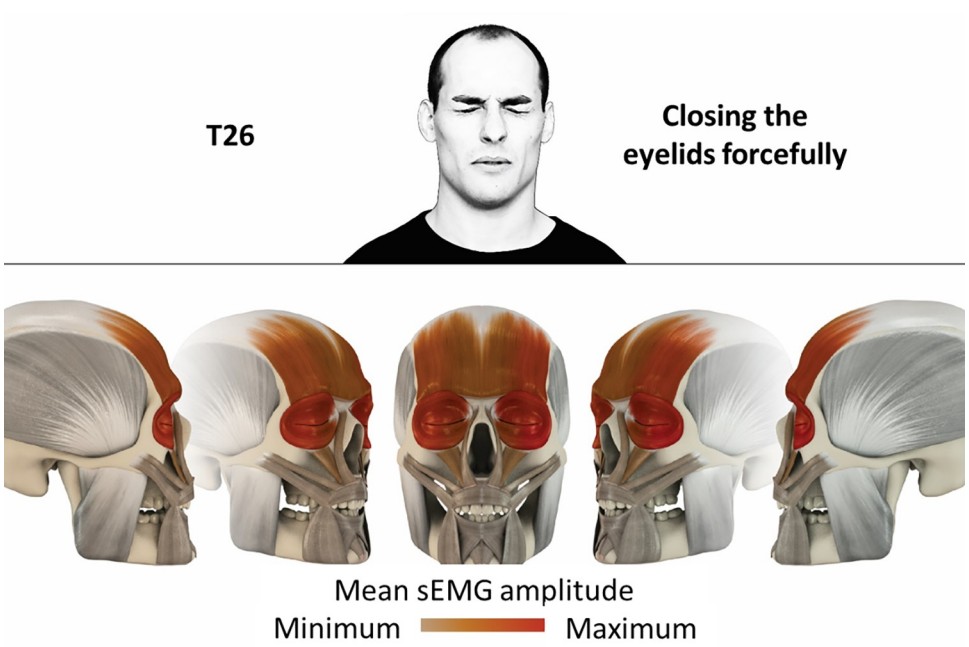

**Fig 27. Color-coded facial muscle activation pattern of task T26.** Closing eyelids forcefully. Animation by Jonas Lauströer.

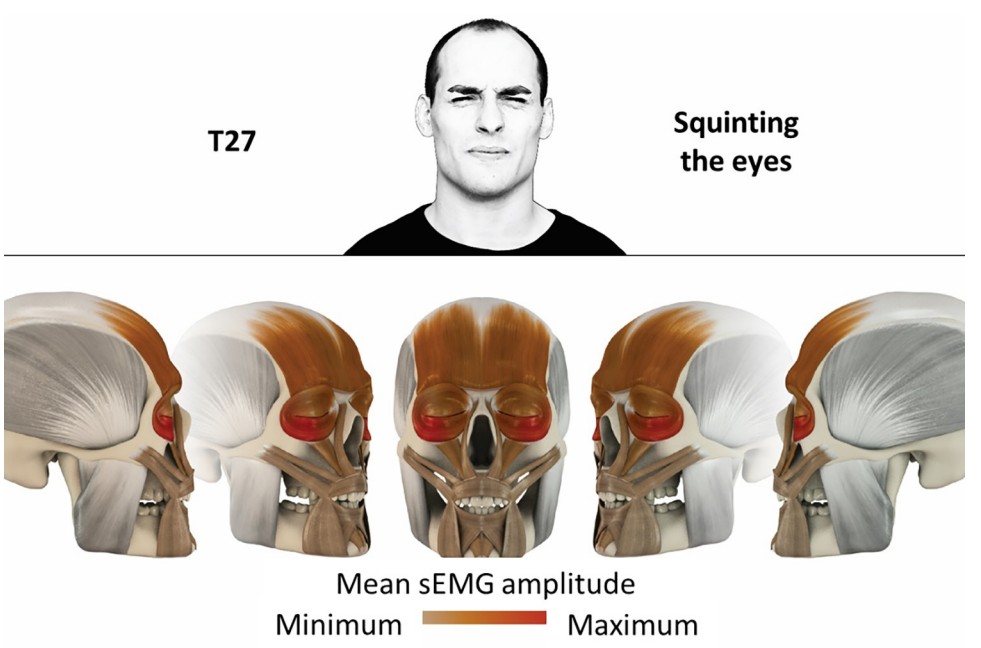

**Fig 28. Color-coded facial muscle activation pattern of task T27.** Squinting the eyes. Animation by Jonas Lauströer.

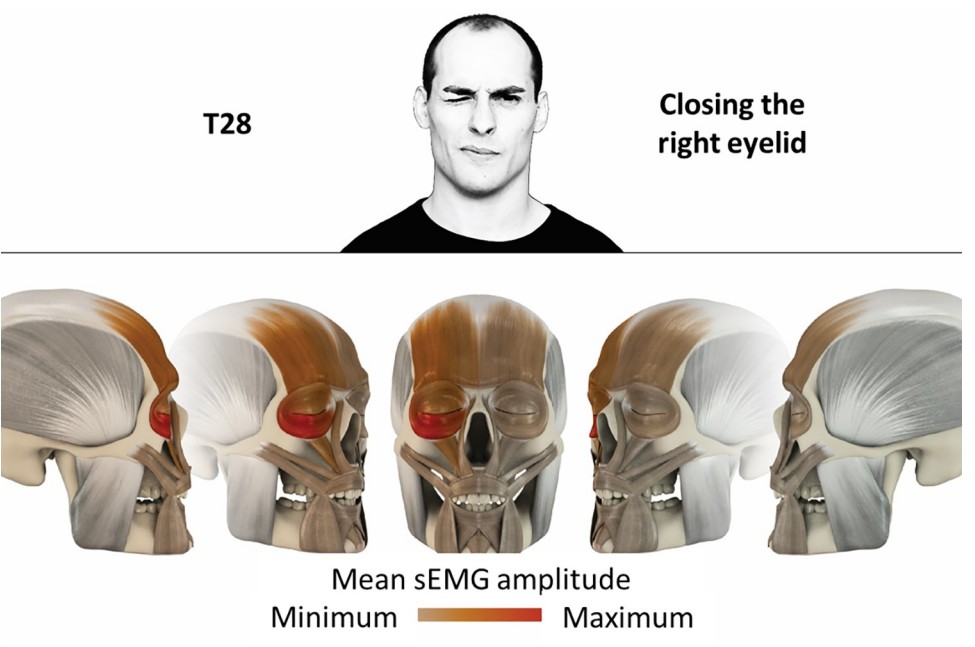

**Fig 29. Color-coded facial muscle activation pattern of task T28.** Closing the right eyelid. Animation by Jonas Lauströer.

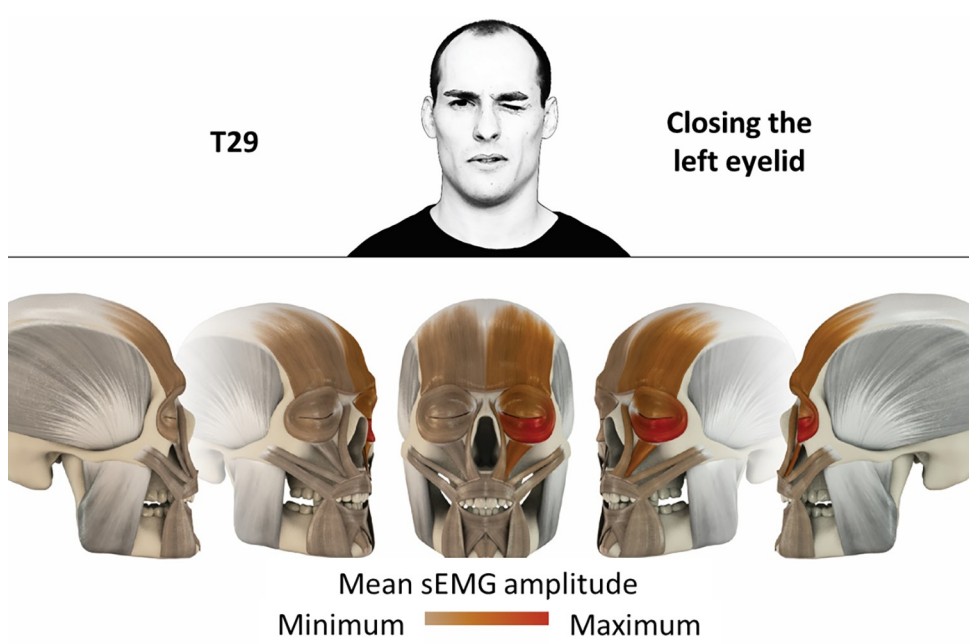

**Fig 30. Color-coded facial muscle activation pattern of task T29.** Closing the left eyelid. Animation by Jonas Lauströer.

## Discussion

The atlas shows objective, quantified and validated activation and coordination patterns of facial muscles based on multi-channel sEMG. The method took into account the main ocular, nasal and oral muscle regions of the face. Using the EMG activity distribution, differences between the various movement tasks important for clinical and psychological studies are illustrated. For such experiments, it is important to measure both sides of the face. Although there is no systematical right and left side difference in healthy probands, there can be an individual right and left side difference in the strength of the activation of a certain facial muscle [9]. Excluding the ear muscles and the platysma, the mimic musculature consists of 19 muscles [11]. We did not record and visualize the following 9 muscles: occipitalis, procerus, corrugator supercilii, depressor supercilii, nasalis, depressor septi nasi, risorius, buccinator, and levator anguli oris. These muscles will be involved in some of the presented facial tasks, too. For instance, the procerus and corrugator supercilii muscle are important muscles for downward eye brow movement. This is of clinical importance, as these two muscles are typical botulinum toxin injection targets for glabellar rhytides [12]. The simultaneous recording was limited to 48 EMG channels. In the future, high-density sEMG will allow to cover a larger area and more muscles with higher resolution [6]. Furthermore, the mimic muscles are relatively small, thin and interwoven. They are partly in layers on top of each other. Since myoelectric potentials spread spatially from their source, crosstalk cannot generally be ruled out in a sEMG registration [13]. In addition, myoelectric potentials emanating from deeper muscles are attenuated significantly more than potential from muscles lying on the surface [14]. For instance, sEMG recording of the corrugator supercilii could also record activity from the orbicularis oculi, levator labii superioris alaeque nasi, or frontalis [15]. Hence, more EMG channels will not automatically lead to a better separation (and visualization in an atlas) of individual muscle

function. Finally, the risk of unnatural obstruction that the electrodes pose to the production of facial expressions will increase with more electrodes. In this study, only monopolar EMG signals were used to derive the facial muscles atlas. The monopolar EMG signal reflects both superficial and deep EMG sources. Bipolar EMG signals only reveal superficial EMG sources [6,16,17]. An additional analysis of bipolar EMG signals might help to better separate superficial from deeper facial muscle activation.

The atlas has two other limitations: The colored images visualize the activation pattern of the facial muscles while the head is upright position. We cannot exclude that these sEMG pattern change in other body postures. This has not been analyzed yet. Furthermore, the analysis was performed only in men. Although facial sEMG activation patterns were generated for men and women in some studies [6,8,17], gender differences were not yet analyzed. Therefore, we cannot rule out that women show other facial muscle expression patterns.

The visualized pattern makes it clear that not only individual muscles contract during a facial movement or task. Every facial movement is carried by a group of muscles. In addition to the agonist, which determines the main direction of the soft tissue displacement within the face through its force vector, synergistic but also antagonistic muscles are activated, which have a modulating and stabilizing effect on facial expressions [1]. Only through the interplay of different facial muscles the mimic expression patterns are made possible in all their diversity. About 30–40% of patients with acute facial paralysis do not recover completely and develop synkinesis [18]. Postparalytic facial synkinesis is a disfiguring condition characterized by involuntary contraction of one or more facial muscles during voluntary movement of other facial muscles [19]. It will be worthwhile to study such patients with altered facial muscle activation pattern, and to compare the data to the atlas to better design rehabilitation programs specifically addressing function relevant combinations of facial muscle activation.

The Facial Action Coding System (FACS) is a standardized method to label facial movements to measure emotional facial expressions [20]. Originally, facial EMG played an important role to define the underlying mimic muscle movements. The activity of specific muscle groups is coded as action units (AU) by human coders or by automated video analysis [21,22]. S1 Table compares the present data to the AU muscular basis of the different tasks. The present atlas does not only cover the important tasks for AU related emotional expression experiments, but add other important tasks for speech experiments and more detailed orofacial tasks for facial rehabilitation in patients with oral dysfunction [23–25]. Hence, the present atlas should become a helpful tool to design sEMG experiments not only for clinical trials and psychological experiments, but also for speech therapy and orofacial rehabilitation studies. For instance, in case of Bell's palsy, the most common course of peripheral facial neuromuscular dysfunction [26], classification of the initial severity of facial dysfunction, its changes after therapy and during follow-up is mainly done by clinical grading scales dependent on the subjective evaluation of the observer [27]. This leads to limited inter-observer and intra-observer reliability. sEMG mapping offers an objective profiling of facial muscle activity [6]. In combination with image analysis of facial movements, this will help to understand the connection between facial muscle activation and changes of the facial surface geometry [6]. This is important to develop reliable tools for automated facial nerve dysfunction classification [28]. Furthermore, sEMG mapping might allow a high-resolution depiction of regional facial muscle impairment dyscoordination in patients with chronic facial nerve dysfunction. This should help to improve planning of individual physiotherapy or even better planning of facial rehabilitation surgery [29,30].

## Conclusions

Overall, the visualized EMG activity distribution patterns presented here give an overview of the activation and coordination patterns of a large number of facial movements. They demonstrate the complexity and diversity of facial-muscular activation processes. In addition, in terms of acute therapy and rehabilitation, they form a goal-oriented approach for conservative treatment strategies and restorative surgical interventions in patients suffering from facial paralysis.

## Supporting information

**S1 Table. Mimic tasks in comparison to action units (AU) of the Facial Action Coding System (FACS).**
(DOCX)

## Acknowledgments

The authors wish to thank Mr. Jonas Lauströer for the excellent illustrations.

## Author Contributions

**Conceptualization:** Nikolaus P. Schumann, Hans C. Scholle, Orlando Guntinas-Lichius.

**Data curation:** Nikolaus P. Schumann, Hans C. Scholle, Orlando Guntinas-Lichius.

**Formal analysis:** Nikolaus P. Schumann, Kevin Bongers, Hans C. Scholle.

**Funding acquisition:** Orlando Guntinas-Lichius.

**Investigation:** Kevin Bongers.

**Methodology:** Nikolaus P. Schumann, Hans C. Scholle, Orlando Guntinas-Lichius.

**Project administration:** Nikolaus P. Schumann.

**Supervision:** Nikolaus P. Schumann, Hans C. Scholle, Orlando Guntinas-Lichius.

**Validation:** Nikolaus P. Schumann, Hans C. Scholle, Orlando Guntinas-Lichius.

**Visualization:** Hans C. Scholle, Orlando Guntinas-Lichius.

**Writing – original draft:** Orlando Guntinas-Lichius.

**Writing – review & editing:** Nikolaus P. Schumann, Kevin Bongers, Hans C. Scholle, Orlando Guntinas-Lichius.

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
