## [Decision Letter · Decision Letter 0]

23 Jun 2021

PONE-D-21-15420

Atlas of voluntary facial muscle activation: Visualization of surface electromyographic activities of facial muscles during mimic exercises

PLOS ONE

Dear Dr. Guntinas-Lichius,

Thank you for submitting your manuscript to PLOS ONE. After careful consideration, we feel that it has merit but does not fully meet PLOS ONE’s publication criteria as it currently stands. Therefore, we invite you to submit a revised version of the manuscript that addresses the points raised during the review process.

We look forward to receiving your revised manuscript.

Kind regards,

Yingchun Zhang, Ph.D

Academic Editor

PLOS ONE

Journal Requirements:

3. We note that Figures 2-30 includes an images of a participant in the study. 

Reviewers' comments:

Reviewer's Responses to Questions

**Comments to the Author**

1. Is the manuscript technically sound, and do the data support the conclusions?

Reviewer #1: Yes

Reviewer #2: Yes

2. Has the statistical analysis been performed appropriately and rigorously? 

Reviewer #1: Yes

Reviewer #2: N/A

3. Have the authors made all data underlying the findings in their manuscript fully available?

Reviewer #1: Yes

Reviewer #2: Yes

4. Is the manuscript presented in an intelligible fashion and written in standard English?

Reviewer #1: Yes

Reviewer #2: Yes

5. Review Comments to the Author

Reviewer #1: This interesting research article presents the derivation of an sEMG derived facial activation atlas. The authors utilized sEMG sensors distributed over twelve facial muscles to derive activation maps during twenty-nine specific facial tasks. The authors present the convincing ability to discern specific regions of facial muscle activity during specific facial tasks. I have several comments and questions regarding the study.

1. The introduction section should be expanded to include some background information on previous attempts to map facial muscle activity using EMG.

2. Why were only male subjects recruited? Do the authors believe these results are generalizable to a wider sex population, and if so why? I see that the authors cite ref [7] as support that male and female facial activation patterns do not differ, but I do not see that reference making such a conclusion.

3. It appears that recordings were made using duplicate electrodes over each target muscle, yet only monopolar recordings were used to derive the atlas. Did the authors explore using bipolar signals to construct or supplement their facial activation atlas? It may be an interesting point to discuss.

4. Although published in great detail in the J Neuroscience Methods paper that this article is based on, some of the necessary details (amplification sampling rate, electrode placement verification, statistical normalization, etc.) should also be reported in this article as well.

5. I believe that the methods section regarding the generation of the atlas needs to be expanded. The implementation of a model with color coded muscles is the major contribution of this paper, but how the model was generated is not sufficiently described. How was the 3D model created? What programming language or tools were used to generate the atlas’ figures?

6. Could the authors further discuss potential applications of the atlas in the discussion section? I can see some use in characterizing the facial activity patients with facial paresis after stroke, or in Bell’s palsy patients.

Reviewer #2: This paper by Guntinas-Lichius et al. organizes and depicts the groups of facial muscles that are activated with certain expressions and with phonation of specific German vowels into a very well-illustrated atlas. The authors used surface EMG to detect activity of contracting facial muscles with certain tasks and generated an atlas of muscle activation corresponding to each task. The function of facial muscles individually are well understood and utilized in plastic surgery for cosmetic botox injections and management of facial asymmetry/synkinesis following facial nerve injury. While the function of these muscles is already well known, the authors have uniquely made an atlas that depicts the interaction of these individual muscles together when creating certain facial expressions and making certain shapes with the mouth for phonation. This information may be useful in future facial nerve studies and speech rehabilitation after stroke/nerve injury.

This manuscript is thorough with excellent illustrations to depict the muscle groups involved in certain activities. The authors are to be commended for their attention to detail utilizing a 48 channel sEMG on the face to faithfully detect muscle movement across the whole face. This is technically challenging as the face is a small surface area and the muscles on the face are small compared to other muscle groups in the extremities. One weakness of the paper (this weakness was mentioned by the authors in the discussion), is that the electrodes were not placed over the depressors of the brow—namely, the corrugator supercilii and procerus muscles. These are important depressors of the brow that are commonly treated with botox injection in cosmetic practices. Because these were not sampled, the only relevant muscle activity identified with contraction of the eyebrows (no. 25 on table 1, figure 26 atlas) was the frontalis, which is a well known elevator (not depressor) of the brow. While I have no doubt there is some frontalis activity with contraction of the eyebrow, I believe the atlas is misleading on this particular facial activity (“contracting the eyebrows”) because the corrugators and procerus were not sampled.

Overall, the authors have been thorough with their evaluation and depiction of facial muscle groups and have identified future studies that may be pursued for future clinical and scientific use.

6. PLOS authors have the option to publish the peer review history of their article (what does this mean?). If published, this will include your full peer review and any attached files.

Reviewer #1: No

Reviewer #2: No

---

## [Author Response · Author response to Decision Letter 0]

27 Jun 2021

Point-by-point response

PONE-D-21-15420: Atlas of voluntary facial muscle activation: Visualization of surface electromyographic activities of facial muscles during mimic exercises

Thank you very much for the detailed reviews. We answer all comments and queries point-by-point.

Editorial team

0.1. Please ensure that your manuscript meets PLOS ONE's style requirements, including those for file naming. The PLOS ONE style templates can be found at

Answer 0.1: We checked again the format, including file naming.

0.2. Please provide additional details regarding participant consent. In the ethics statement in the Methods and online submission information, please ensure that you have specified what type you obtained (for instance, written or verbal, and if verbal, how it was documented and witnessed). If your study included minors, state whether you obtained consent from parents or guardians. If the need for consent was waived by the ethics committee, please include this information.

Answer 0.2: The ethics statement in the Methods on page 4 and the online submission contain all mentioned information.

0.3. We note that Figures 2-30 includes an images of a participant in the study. 

As per the PLOS ONE policy (http://journals.plos.org/plosone/s/submission-guidelines#loc-human-subjects-research) on papers that include identifying, or potentially identifying, information, the individual(s) or parent(s)/guardian(s) must be informed of the terms of the PLOS open-access (CC-BY) license and provide specific permission for publication of these details under the terms of this license. Please download the Consent Form for Publication in a PLOS Journal (http://journals.plos.org/plosone/s/file?id=8ce6/plos-consent-form-english.pdf). The signed consent form should not be submitted with the manuscript, but should be securely filed in the individual's case notes. Please amend the methods section and ethics statement of the manuscript to explicitly state that the patient/participant has provided consent for publication: “The individual in this manuscript has given written informed consent (as outlined in PLOS consent form) to publish these case details”. If you are unable to obtain consent from the subject of the photograph, you will need to remove the figure and any other textual identifying information or case descriptions for this individual.

Answer 0.3: We have the consent from the participant. We upload the form. We added in the Methods on page 4 this sentence: “The individual shown in the figures of this manuscript has given written informed consent to publish these case details.”.

Reviewer #1:

This interesting research article presents the derivation of a sEMG derived facial activation atlas. The authors utilized sEMG sensors distributed over twelve facial muscles to derive activation maps during twenty-nine specific facial tasks. The authors present the convincing ability to discern specific regions of facial muscle activity during specific facial tasks. I have several comments and questions regarding the study.

1.1. The introduction section should be expanded to include some background information on previous attempts to map facial muscle activity using EMG.

Answer 1.1: Done. In the Introduction on pages 3-4 we added further information on the attempts and strategies to map facial muscle activity using EMG.

1.2. Why were only male subjects recruited? Do the authors believe these results are generalizable to a wider sex population, and if so why? I see that the authors cite ref [7] as support that male and female facial activation patterns do not differ, but I do not see that reference making such a conclusion.

Answer 1.2: Yes, the wording was too imprecise. Ref. 7 (in the first version of the manuscript, numbers have changed now) analyzed male and female probands, the data in the paper show no difference, but the paper does not include a formal statistical comparison of both genders. Therefore, we changed the wording in the Discussion on page 13. Now we clearly state that we cannot rule out that women have other expression patterns.

1.3. It appears that recordings were made using duplicate electrodes over each target muscle, yet only monopolar recordings were used to derive the atlas. Did the authors explore using bipolar signals to construct or supplement their facial activation atlas? It may be an interesting point to discuss.

Answer 1.3: Yes. Only monopolar recordings were performed. As recommended in query 1.4, we added more details concerning the recording in the Methods, on page 5, see Answer 1.4 below. No, we did not explore the use of bipolar signals. We added this aspect on page 13 of the Discussion: “In this study, only monopolar EMG signals were used to derive the facial muscles atlas. The monopolar EMG signal reflects both superficial and deep EMG sources. Bipolar EMG signals only reveal superficial EMG sources […]. An additional analysis of bipolar EMG signals might help to better separate superficial from deeper facial muscle activation.”

1.4. Although published in great detail in the J Neuroscience Methods paper that this article is based on, some of the necessary details (amplification sampling rate, electrode placement verification, statistical normalization, etc.) should also be reported in this article as well.

Answer 1.4: We added some necessary details on page 5 in the Methods now. We tried to avoid a complete repetition of the detailed method description given in the cited J Neuroscience Methods paper.

1.5. I believe that the methods section regarding the generation of the atlas needs to be expanded. The implementation of a model with color coded muscles is the major contribution of this paper, but how the model was generated is not sufficiently described. How was the 3D model created? What programming language or tools were used to generate the atlas’ figures?

Answer 1.5: Done. We explained now in the Methods on page 6 the 3D visualization in more detail. 

1.6. Could the authors further discuss potential applications of the atlas in the discussion section? I can see some use in characterizing the facial activity patients with facial paresis after stroke, or in Bell’s palsy patients.

Answer 1.6: At the end of the Discussion on page 14 we extended the part on potential applications: “For instance, in case of Bell’s palsy, the most common course of peripheral facial neuromuscular dysfunction […], classification of the initial severity of facial dysfunction, its changes after therapy and during follow-up is mainly done by clinical grading scales dependent on the subjective evaluation of the observer […]. This leads to limited inter-observer and intra-observer reliability. sEMG mapping offers an objective profiling of facial muscle activity […]. In combination with image analysis of facial movements, this will help to understand the connection between facial muscle activation and changes of the facial surface geometry […]. This is important to develop reliable tools for automated facial nerve dysfunction classification […]. Furthermore, sEMG mapping might allow a high-resolution depiction of regional facial muscle impairment dyscoordination in patients with chronic facial nerve dysfunction. This should help to improve planning of individual physiotherapy or even better planning of facial rehabilitation surgery{[…].”.

Reviewer #2:

This paper by Guntinas-Lichius et al. organizes and depicts the groups of facial muscles that are activated with certain expressions and with phonation of specific German vowels into a very well-illustrated atlas. The authors used surface EMG to detect activity of contracting facial muscles with certain tasks and generated an atlas of muscle activation corresponding to each task. The function of facial muscles individually are well understood and utilized in plastic surgery for cosmetic botox injections and management of facial asymmetry/synkinesis following facial nerve injury. While the function of these muscles is already well known, the authors have uniquely made an atlas that depicts the interaction of these individual muscles together when creating certain facial expressions and making certain shapes with the mouth for phonation. This information may be useful in future facial nerve studies and speech rehabilitation after stroke/nerve injury.

This manuscript is thorough with excellent illustrations to depict the muscle groups involved in certain activities. The authors are to be commended for their attention to detail utilizing a 48 channel sEMG on the face to faithfully detect muscle movement across the whole face. This is technically challenging as the face is a small surface area and the muscles on the face are small compared to other muscle groups in the extremities.

2.1. One weakness of the paper (this weakness was mentioned by the authors in the discussion), is that the electrodes were not placed over the depressors of the brow—namely, the corrugator supercilii and procerus muscles. These are important depressors of the brow that are commonly treated with botox injection in cosmetic practices. Because these were not sampled, the only relevant muscle activity identified with contraction of the eyebrows (no. 25 on table 1, figure 26 atlas) was the frontalis, which is a well known elevator (not depressor) of the brow. While I have no doubt there is some frontalis activity with contraction of the eyebrow, I believe the atlas is misleading on this particular facial activity (“contracting the eyebrows”) because the corrugators and procerus were not sampled. Overall, the authors have been thorough with their evaluation and depiction of facial muscle groups and have identified future studies that may be pursued for future clinical and scientific use.

Answer 2.1: We added more on this limitation in the Discussion on page 12: “These muscles will be involved in some of the presented facial tasks, too. For instance, the procerus and corrugator supercilii muscle are important muscles for downward eye brow movement. This is of clinical importance, as these two muscles are typical botulinum toxin injection targets for glabellar rhytides […]”

Orlando Guntinas-Lichius

For all co-authors

Jena, 27-June-2021

---

## [Decision Letter · Decision Letter 1]

7 Jul 2021

Atlas of voluntary facial muscle activation: Visualization of surface electromyographic activities of facial muscles during mimic exercises

PONE-D-21-15420R1

Dear Dr. Guntinas-Lichius,

We’re pleased to inform you that your manuscript has been judged scientifically suitable for publication and will be formally accepted for publication once it meets all outstanding technical requirements.

Kind regards,

Yingchun Zhang, Ph.D

Academic Editor

PLOS ONE

Additional Editor Comments (optional):

Reviewers' comments:

Reviewer's Responses to Questions

**Comments to the Author**

1. If the authors have adequately addressed your comments raised in a previous round of review and you feel that this manuscript is now acceptable for publication, you may indicate that here to bypass the “Comments to the Author” section, enter your conflict of interest statement in the “Confidential to Editor” section, and submit your "Accept" recommendation.

Reviewer #1: All comments have been addressed

Reviewer #2: All comments have been addressed

2. Is the manuscript technically sound, and do the data support the conclusions?

Reviewer #1: Yes

Reviewer #2: Yes

3. Has the statistical analysis been performed appropriately and rigorously? 

Reviewer #1: Yes

Reviewer #2: N/A

4. Have the authors made all data underlying the findings in their manuscript fully available?

Reviewer #1: Yes

Reviewer #2: Yes

5. Is the manuscript presented in an intelligible fashion and written in standard English?

Reviewer #1: Yes

Reviewer #2: Yes

6. Review Comments to the Author

Reviewer #1: (No Response)

Reviewer #2: The authors have addressed the primary concern with additional explanation in lines 220-225 of the manuscript. I have no further reservations.

7. PLOS authors have the option to publish the peer review history of their article (what does this mean?). If published, this will include your full peer review and any attached files.

Reviewer #1: No

Reviewer #2: No

---

## [Editor Report · Acceptance letter]

9 Jul 2021

PONE-D-21-15420R1 

Atlas of voluntary facial muscle activation: Visualization of surface electromyographic activities of facial muscles during mimic exercises 

Dear Dr. Guntinas-Lichius:

I'm pleased to inform you that your manuscript has been deemed suitable for publication in PLOS ONE. Congratulations! Your manuscript is now with our production department. 

Kind regards, 

on behalf of

Dr. Yingchun Zhang 

Academic Editor

PLOS ONE